# Citrus Waste as Source of Bioactive Compounds: Extraction and Utilization in Health and Food Industry

**DOI:** 10.3390/molecules28041636

**Published:** 2023-02-08

**Authors:** Zahra Maqbool, Waseem Khalid, Hafiz Taimoor Atiq, Hyrije Koraqi, Zaryab Javaid, Sadeq K. Alhag, Laila A. Al-Shuraym, D. M. D. Bader, Mohammed Almarzuq, Mohamed Afifi, Ammar AL-Farga

**Affiliations:** 1Department of Food Science, Faculty of Life Sciences, Government College University, Faisalabad 38000, Pakistan; 2Department of Food Science and Technology, Muhammad Nawaz Sharif University of Agriculture, Multan 23546, Pakistan; 3Faculty of Food Science and Biotechnology, UBT-Higher Education Institution, Rexhep Krasniqi No. 56, 10000 Pristina, Kosovo; 4Department of Pharmacy, University of Central Punjab, Lahore 54590, Pakistan; 5Biology Department, College of Science and Arts, King Khalid University, Muhayl Asser 61913, Saudi Arabia; 6Biology Department, Faculty of Science, Princess Nourah Bint Abdulrahman University, Riyadh 11671, Saudi Arabia; 7Chemistry Department, College of Science, King Khalid University, P.O. Box 9004, Abha 61413, Saudi Arabia; 8Unit of Scientific Research, Applied College, Qassim University, Buraidah 52571, Saudi Arabia; 9Biochemistry Department, Faculty of Sciences, University of Jeddah, Jeddah 21959, Saudi Arabia; 10Department of Biochemistry, Faculty of Veterinary Medicine, Zagazig University, Zagazig 44519, Egypt; 11Najla Bint Saud Al Saud Center for Distinguished Research in Biotechnology, Jeddah 21577, Saudi Arabia

**Keywords:** citrus waste, potential health benefits, bioactive compounds, EOs

## Abstract

The current research was conducted to extract the bioactive compounds from citrus waste and assess their role in the development of functional foods to treat different disorders. The scientific name of citrus is *Citrus* L. and it belongs to the *Rutaceae* family. It is one of the most important fruit crops that is grown throughout the world. During processing, a large amount of waste is produced from citrus fruits in the form of peel, seeds, and pomace. Every year, the citrus processing industry creates a large amount of waste. The citrus waste is composed of highly bioactive substances and phytochemicals, including essential oils (EOs), ascorbic acid, sugars, carotenoids, flavonoids, dietary fiber, polyphenols, and a range of trace elements. These valuable compounds are used to develop functional foods, including baked products, beverages, meat products, and dairy products. Moreover, these functional foods play an important role in treating various disorders, including anti-aging, anti-mutagenic, antidiabetic, anti-carcinogenic, anti-allergenic, anti-oxidative, anti-inflammatory, neuroprotective, and cardiovascular-protective activity. EOs are complex and contain several naturally occurring bioactive compounds that are frequently used as the best substitutes in the food industry. Citrus essential oils have many uses in the packaging and food safety industries. They can also be used as an alternative preservative to extend the shelf lives of different food products.

## 1. Introduction

Citrus is one of the most important fruit crops in the world. The scientific name of citrus is *Citrus* L. and it is a member of the *Rutaceae* family. It is grown extensively in areas that are classified as tropical or subtropical, as well as many other areas, which together produce more than 100 million tons annually [1]. Customers display a significant preference for citrus fruits because of their attractive colors, flavorful aromas, and pleasant flavors. Worldwide, citrus fruits are most commonly cultivated crops. Citrus fruits are an important staple food in the diets of people all over the world [2]. These fruits are crucial throughout the world both nutritionally and commercially [3]. Citrus species could be a source of valuable oils that could be used in food and for other industrial purposes [4]. Around the world, citrus fruits are considered nutrient-dense, energy-dense, and health-promoting fruits. Several of these fruits (lemon, grapefruit, sweet orange, citron, clementine, and pomelo) have also been used as traditional medicinal herbs in Asian countries to treat a variety of illnesses. Several studies showed that citrus fruits contained secondary metabolites as well as bioactive compounds that can be used as either chemotherapeutics or supplements [5,6]. The secondary metabolites found in citrus fruits are important to human health due to their functional properties. There are different secondary metabolites present in citrus waste, including coumarins, alkaloids, carotenoids, limonoids, phenolic acids, flavonoids, and essential oils (EOs) [7]. The utilization of plant residues can reduce the risk of metabolic syndrome-related ailments such as neurodegenerative diseases, diabetes, cardiovascular disease, and cancer. *Citrus* L. is a basic fruit crop that contains high levels of flavonoids, carotenoids, limonoids, terpenes, and other bioactive components [8]. Citrus fruits are abundant in vitamins C, A, and E, minerals, coumarins, flavonoids, limonoids, pectins, carotenoids, and other beneficial phytochemicals. The phytochemicals are consumed in the form of fresh fruits or products and exert positive effects on people’s health, including anti-mutagenicity, antioxidant, anti-carcinogenic, anti-inflammatory, and anti-aging effects [9]. Additionally, these phytochemicals promote cardiovascular health and nervous system function [7]. Citrus fruit phytochemicals may have antioxidant properties via raising liver protective enzyme activity, blocking lipids to prevent deoxyribonucleic acid (DNA) damage, and bolstering the immune system [10]. Citrus fruits are prized for their nutritional content and for providing some of the world’s most popular flavors. Growing citrus fruits (tangerines, lemons, oranges, grapefruits, and limes) is one of the most common means of producing fruit across the globe. Due to expanding customer demands, their output is increasing year after year [11]. Every year, the citrus processing industry produces a large amount of fruit waste, with citrus peel waste accounting for over half of the wet fruit mass. The waste from citrus has fundamental economic value because it contains a great deal of EOs, ascorbic acid, sugars, carotenoids, flavonoids, dietary fiber, polyphenols, and trace elements. This waste also includes a high concentration of sugars that can be fermented to produce bioethanol. These components are useful in the production of medicines, cosmetics, and food supplements [12].

## 2. Different Sources of Citrus Waste

Oranges, lemons, grapefruits, mandarins, and limes are the citrus fruits that are most commonly grown around the world. Due to escalating consumer demands, production grows yearly. Annually, the industries that process citrus produce large amounts of waste, with citrus peel waste alone representing nearly 50% of the wet fruit mass [13]. Food waste is composed of ingredients that are useful for human utilization, but these are degraded, polluted, and discarded. Food waste is an increasing issue that affects all facets of waste management. In order to develop long-term solutions, all actors involved in the food supply chain, including the industry, agriculture, merchants, collection to disposal processes, and consumers, must participate [14]. The processing of citrus fruit generates a range of waste, including liquid, solid, and distillery effluents. Rags, peels, sludge, seeds, and residue constitute solid waste. However, there are different types of liquid waste too, including cannery effluents, can cooler overflows, fruit washing wastewaters, sectioning table and peeling wastewaters, and floor flushing water [15]. Citrus is primarily used in the food manufacturing process to create fresh juice or drinks with citrus flavors, resulting in a significant amount of citrus waste each year in the form of peels, pulp, and seeds. However, the seeds, pulp, and peels of many fruits and vegetables may contain a significant number of bioactive compounds [16]. A previous study was conducted on the recovery of vitamin C, phenolic compounds, and antioxidant activity from citrus (lemon, orange, and grapefruit) fruit waste. Each citrus fruit’s peel, whole fruit, and pulp with seeds were converted into ethanol extracts. Results showed that peels had higher levels of phenolic compounds, vitamin C, flavonoids, and antioxidant activity than their internal, wasted parts (seeds and pulp) in each citrus variety [17]. The extracts (peel, pulp, and seeds) of *Citrus reticulata* (Phlegraean mandarin), *Citrus japonica* (Kumquat), and *Citrus clementina* were compared and characterized in terms of photosynthetic pigment content, total polyphenol amount, antioxidant activity, and vitamin C [18]. The generation of waste from citrus processing is shown in Figure 1.

### 2.1. Citrus Peel

In the quest to reduce global food losses and waste, citrus peel waste (CPW) in particular has emerged as a sustainable and promising option for biorefinery without competing with human foods and animal feeds. According to recent studies, CPW is widely produced and has the industrial potential to be biologically converted into fuels and chemicals [19]. The extraction of essential oils (limonene) and pectin for use as cosmetic and food additive ingredients is the most widely known method for using CPW [20]. Fruit waste is produced in large quantities by agricultural processes all over the world. Frequently, this waste is simply discarded into landfills or the ocean. Fruit waste contains a variety of sugars, such as fructose, sucrose, and glucose, which can be fermented to produce bioethanol. Some fruit wastes, such as citrus peel waste (CPW), contain substances that can stop fermentation, which is necessary for efficient bioethanol production. A novel method for converting CPW from a single source (mandarin, orange, lemon, grapefruit, or lime) or CPW combined with other fruit waste (apple pomace, banana peel, or pear waste) into bioethanol has been developed [21].

### 2.2. Citrus Seeds

After the seeds are gathered, cleaned, dried, and ground, it is necessary to optimize a number of downstream processes in order to extract phytoconstituents from them. Solvents, time, temperature, pressure, and particle size are some of the steps involved in downstream processes. Various extraction techniques are used for citrus seeds, including supercritical CO_2_ extraction, solvent extraction, cold pressing, and ultrasound-assisted extraction [22]. Citrus seeds are typically discarded as waste because they are thought to be useless. According to Ammerman and Arrington [23], the average percentage of seeds in dried citrus pulp is 4.8%. These discarded seeds can be used profitably as a protein supplement for livestock, because they are high in protein. Furthermore, citrus seeds have high potential for use as biodiesel due to their 30% oil content (by weight). According to a rough analysis of flour produced from unhulled and dehulled citrus seeds, it contained 28.5% carbohydrates, 52% fat, 3.1% crude protein, 5.5% crude fiber, and 2.5% ash (dry basis). Rashid et al. [24] produced methyl esters that complied with both the ASTM D6751 and EN 14214 biodiesel standards by trans-esterifying citrus seed oil with methanol under the catalysis of sodium methoxide.

### 2.3. Citrus Pomace

Citrus pomace is the leftover residue from the processing of citrus fruit to create juice or other products. When citrus fruits are processed into industrial citrus juice, nearly half of their volume is wasted, and the enormous amount of citrus pomace causes serious ecological problems [25]. In the case of not using adequate citrus, the pomace generated by the agro-fruit industry has a negative impact on the environment and will result in significant financial losses. Fruit pomaces are rich in a variety of beneficial bioactive substances, including dietary fiber, carbohydrates, phenolic compounds, polysaccharides, phytochemicals, natural antioxidants, and a number of other nutrients that are beneficial for health [26]. In a previous study, the effect of various solvents was examined in relation to the recovery of vitamin C, flavonoids, phenolic compounds, and extractable solids (ES) from lemon pomace waste. The results showed the effects of various solvents, such as water, methanol, ethanol, and acetone. The total phenolic compounds, vitamin C, total flavonoids, and antioxidant activity of lemon pomace were measured by using a combination of solvents [27]. Moisture (10%), fat (0.4–1.6%), sugars (30–40%), pectin (14–25%), micronutrients (0.5%), and hemicellulose and cellulose (13–17%) constitute the bulk of dried citrus pomace [28].

## 3. Extraction of Bioactive Compounds from Citrus Waste

Culturing citrus fruits requires extensive processing. However, after processing, a number of by-products and wastes are produced that are rich sources of bioactive components, such as pectin, essential oils (EOs), and water-insoluble and -soluble antioxidants. The major phytochemicals of *Citrus* L. waste are shown in Figure 2. While some of these wastes are now valorized in various ways, such as non-toxic and effective methods, different techniques used for successful extraction might fundamentally improve their valorization and produce greater revenues and high-quality bioactives [29].

The by-products of citrus and their waste include significant amounts of substances of high value, and they can be utilized for a variety of technological and health-related purposes, as mentioned in Table 1. Citrus by-products contain bioactive compounds with biological activity, including carotenoids, polyphenols, and essential oils (EOs). Carotenoids and polyphenols have numerous health benefits that are related to their antioxidant activity [30]. Citrus fruits are frequently processed into turbid juices, and the residue part is considered waste, including juice vesicles, membranes, peels, and seeds. These account for 45 to 60% of the weight discarded [31]. Due to the abundance of polyphenols, EOs, flavonoids, dietary fiber, sugars, ascorbic acid, and carotenoids in orange peel, it can be utilized as a source of economically useful, elevated components, in areas such as solid biofuel, bioabsorbents, animal feed, fertilizer, EOs, pectin, ethanol, methane, industrial enzymes, and single-cell protein [32,33,34,35].

### 3.1. Pectin

Ngouémazong et al. [42] described pectin as an emulsifier, texturizer, thickener, and stabilizer in food. It is utilized in a variety of applications, including fillings for confectionery and dietary fiber supplements [43]. It is derived commercially from the peels of orange, lime, lemon, and grapefruit as a white to pale brown powder [44]. The extraction of pectin is among the most efficient and cost-effective methods [45]. In a factory setting, citrus peels and rinds are heated to approximately 100 °C and subjected to acidic conditions to extract pectin [46]. Alternative extraction methods for pectin extraction have recently been developed, including ultrasonic extraction (USE) [47], microwave-assisted extraction (MAE) [48], and enzymatic extraction [49]. The hydrolysis of lignocellulosic materials has been greatly aided by the application of subcritical water extraction [50], in addition to processing any remaining orange peel for its pectin [51]. In addition to its technological capabilities in a variety of commodities, pectin can play a role as a dietary fiber and bioactive compound and act as an anti-cancer agent [52] (Figure 3).

### 3.2. Dietary Fiber

Citrus fruits contain both soluble and insoluble fiber that may be separated. Soluble dietary fiber is largely made up of mucus, gum, and pectin; in contrast, hemicellulose, cellulose, and lignin make up the bulk of insoluble dietary fiber [53]. Approximately 50–60% of its dry weight can be attributed to cellulose and hemicellulose, which make it a prime source for both substances. Citrus fiber is a biologically active component (BAC) that contains polyphenol-like components and plays a role as a lipid oxidation inhibitor in meat products, prolonging the stability of the meat and enhancing the overall oxidative stability [54]. Because of their high water and oil absorption rates, the pulp, seeds, and even the peel of oranges have been utilized as a fat substitute in ice cream [55]. Citrus peel is subjected to dietary fiber extraction using subcritical water. Pectin, hemicellulose, and cellulose, together with other dietary fiber sources, were successfully extracted from orange peel through the utilization of a hydrothermal treatment that was carried out at temperatures ranging from 160 to 320 degrees Celsius in a moderate flow extractor. This is an important sequential process for the recovery of useful compounds from citrus fruit waste in a manner that is non-toxic to the environment [56]. 

### 3.3. Essential Oils (EOs)

Plants create essential oils, which are volatile aromatic compounds and are referred to as EOs. These chemicals have been employed as flavoring agents in food, medicine, and cosmetics since ancient times [57]. Citrus species have garnered a great deal of attention due to their abundance of EOs. EOs are highly adaptable and may be used to impart flavor to a large number of products, including beverages, cosmetics, soaps, and household goods [58]. EOs have antibacterial, antifungal, and insecticidal properties, as well as antifungal, antibacterial, and insecticidal activity [59]. Citrus essential oils were extracted from five citrus species by using hydrodistillation. GC–MS analysis showed different phytochemicals in all five citrus oils, including alkaloids, tannins, sterols, terpenoids, saponins, and limonene [60]. A previous study investigated the antioxidant properties, antimicrobial activity, and chemical composition of the essential oil isolated from the aerial parts of Citrus aurantifolia L. The essential oil was extracted using hydrodistillation. Gas chromatography–mass spectrometry (GC–MS) was used to identify and quantify the chemical constituents of the oil. Thirty-three chemical compounds were identified, in which d-limonene was the major constituent. The minor constituents were 3,7-dimethyl-2,6-octadien-1-ol, geraniol, E-citral, Z-citral, and β-ocimene [61]. The GC–MS analysis of *Citrus aurantium* L. (CAEO) identified at least ten compounds, with 2-β pinene, δ-3 carene, and D-limonene as the major compounds [62]. The current study was conducted on *Citrus medica* limonum, in which leaves’ essential oil was extracted by hydrodistillation and the chemical constituents were analyzed by gas chromatography–mass spectroscopy (GC–MS) and Fourier transform infrared (FTIR) spectroscopy. Eleven components were identified in the leaf oil by GC–MS, in which the major constituents were citronellal, limonene, (E)-citral, 1,6-octadien-3-ol, and 3,7-dimethyl [63].

### 3.4. Carotenoids

Carotenoids are also known as carotenes, which are lipid-soluble hydrocarbons, while xanthophylls are their oxygenated derivatives. Carotenes are found in most red and yellow fruits, green leaves, and many roots [64]. Carotenoids are plentiful in citrus fruits. Carotenoids (zeaxanthin, lutein, and lycopene), cryptoxanthin, and pro-vitamin are found in the highest concentrations in fruits and vegetables. Carotenoids play an important role in human health. They act as antioxidants and aid in the growth and health of bones. They also exert a positive influence on the immune system, increase cell-to-cell communication, improve eye health, and reduce cancer risks. A previous study demonstrated that carotenoids have many health advantages. The capability of carotenoids to generate vitamin A makes them the only well-established healthier alternative to carotenoids in humans [65]. Citrus fruits are a significant source of dietary nutrients due to their high levels of carotenoids. The two primary forms of carotenoids are hydrocarbon carotenoids and oxygenated carotenoids [66]. Methanol and diethyl ether were shown to be excellent solvents to extract carotenoids from sour oranges [67].

### 3.5. Polyphenols

The waste of citrus fruits can be used after management in the proper way. The peels offer a superior source of naturally occurring flavonoids and polyphenolics [68]. There are six distinct types of flavonoids found in peels concerning their chemical structures, including flavones, flavanols, flavanones, anthocyanidins, isoflavones, and flavonols [69]. Flavonoids are phenolic compounds with biological characteristics such as antiallergenic, anti-inflammatory, antiviral, and vasodilation [70]. Flavonoids can protect cells from injury by directly scavenging free radicals and preventing their harmful effects [71]. To obtain polyphenols from the peels of a variety of citrus fruits, a strategy was employed that combined solid and liquid extraction and was founded on an ethanolic solution in water. This method proved to be quick, sustainable, and likely economical, with minimal equipment requirements and ease of operation [72].

## 4. Application of Citrus Waste in the Food Industry

Different studies showed that citrus peels contain essential oils that can be used for multiple purposes in food preservation, food safety, and nutraceutical industries. Various research has produced essential oil (EOs)-based thin films, microencapsulation utilizing nanoemulsion coatings, biodegradable polymers, spray applications, and the antibacterial action mechanism of the active chemicals present in EOs [73]. Cosmetics, medicines, food formulations, and packaging are a few of the industries that use citrus essential oils (CEOs). CEOs have extensive uses in the areas of food safety, packaging, and storage [74]. Figure 4 describes the potential applications of *Citrus* L.

### 4.1. Essential Oil-Containing Polysaccharide-Based Edible Films and Coatings

Food industries face a major challenge in the form of waste, which has to be managed and utilized for multiple purposes. Most of the food industries and food scientists use these food wastes as coating materials to protect food products from various environmental factors. Therefore, food coatings and films have gained wide interest due to their excellent palatability and biodegradability, and their ability to maintain food freshness during storage. In the last decade, polysaccharides have been investigated as a viable material for edible films. Applications related to microbiological food safety demonstrated that these polymeric materials have enormous untapped potential. In a study, essential oils (EOs) were mixed with polysaccharide matrices to further enhance the functional qualities of edible films [75]. In another research work by Perdones et al. [76], cold-stored strawberry was coated with chitosan and lemon essential oil and stored in at 5 °C. Chitosan coatings showed no significant effect on the acidity, pH, and soluble solid content of strawberries throughout the storage period. In the food processing industry, the EOs of citrus lemon peel can be used as a natural antibacterial and antioxidant agent. Lemon peel EOs have antibacterial effectiveness against several food-borne diseases [77].

### 4.2. Application of Fiber Concentrate in Bakery Products

Romero-Lopez et al. [78] conducted a study on the addition of citrus fiber to muffins. A dietary fiber-rich orange bagasse product (DFROBP) boosted the fiber content of the experimental muffins by 40 and 63% relative to a control muffin. By partially substituting wheat flour with DFROBP, it was possible to create baked goods with high levels of total dietary fiber and a high indigestible fraction, dietary traits that may be advantageous for people with varying nutritional requirements [79].

### 4.3. Application of Citrus Pectin in Dairy Drinks

Citrus pectin, a polysaccharide found in plant cell walls, is primarily used in the food processing industry as a stabilizer, thickener, and gelling agent. Pectin can be extracted from the peels of various fruits and their waste; it is used in food industries and is extracted from citrus peels and apple pomace. Pectin is commonly used as a stabilizer agent in acidified milk beverages and yogurts. Acidified milk drinks are very popular due to their natural taste and high nutritional value. Many research works have shown that protein flocculation and whey separation in acidified milk beverages are due to the absence of a stabilizer agent. Furthermore, citrus pectin can be added as a protective colloid to stop this behavior and stabilize milk drinks [80].

### 4.4. Utilization of Citrus Waste in the Beverage Industry

Utilizing citrus waste in the food sector is vital due to its bio-functional properties and its potential as a useful approach to improving food quality and customer health. It contributes to the creation of nutrient-rich foods and beverages of superior quality. Citrus waste contains a high level of vitamin C, and it can be used to produce juices and other nutrient-rich beverages [81]. Citrus waste also contains essential oils (EOs). These EOs can be used in a variety of beverages, food additives, and pharmaceutical, food, and cosmetic products, because they can provide flavor and impart a sweet aroma (fragrance). Due to the abundance of phenols and carotenoids in citrus manufacturing by-products, these materials can be put to good use as biofertilizers to increase the freshness and longevity of food and drink [82].

### 4.5. Applications of Essential Oils (EOs) for Food Safety

For a very long time, EOs have been used in food processing and preservation industries for multiple purposes. Recently, researchers and scientists have focused on antibacterial packaging for food products, edible thin films, effective antibacterial films, nanoemulsions for the preservation of fruits and vegetables, soda/citrus concentrate components, flavoring agents in carbonated colas, soft drinks, and the preservation of meat, fish, and shellfish [73]. EOs are more complex than other types of oils because they contain a variety of naturally occurring bioactive chemicals that are frequently utilized in the food industry as superior substitutes. The antioxidant, antibacterial, and antifungal properties of EOs have been investigated in many research works. EOs obtained from tea tree oil, cinnamon oil, sesame oil, clove oil, lemon oil, chia oil, and thyme oil are only a few examples of oils that have significantly increased antioxidant and antibacterial activity, along with increased cereal storage life and enhanced food security [83]. The key EO groups, such as terpenes and aromatic volatile compounds, contribute to food safety without compromising quality. Because of their varied actions, including antibacterial and antioxidant capabilities, EOs could replace some chemical preservatives, extending the freshness of foods including cereals and vegetables [74].

## 5. Herb–Drug Interaction of *Citrus* L.

Herb–drug interactions (HDIs), a growing concern in the clinical use of conventional drugs, have become more prevalent as traditional herbal medicines are used more frequently. Complex HDIs are linked to complex chemical compositions and multiple potential bioactivities [84,85]. HDIs can be divided into pharmacokinetic interactions and pharmacodynamic interactions based on various interaction pathways, with the former being the focus of earlier studies that focused on drug transport and metabolism. On the one hand, HDIs may affect drug levels and/or activity, which may result in therapeutic failure or unfavorable reactions; on the other hand, some HDIs may have positive clinical effects, such as increased efficacy and decreased side effects [86]. Plant preparations are used as medicines by numerous people all over the world. There is rising interest in and use of phytomedicines, even in industrialized regions such as Europe, where most patients can access conventional therapies. Plant preparations are combined with conventional medications, in addition to being used as an alternative form of therapy [85]. The complex mixtures of bioactive compounds in plants indicate potential for interactions, so these combinations merit careful consideration [87]. Nearly 25% of American adults state that they take a prescription drug and a dietary supplement at the same time. The most well-known example may be St. John’s wort’s induction of CYP enzymes and pGP, but there are many other factors to take into account [78,88]. Health professionals and researchers working on drug discovery need to be more aware of HDI. The potential for HDIs should always be evaluated in the non-clinical safety assessment stage of the drug development process, given the rise in plant-sourced pharmacological actives [89]. The phytochemicals of *Citrus* L. also act as agonists or antagonists with various nuclear receptors, such as PXR, CAR, and AhR, and regulate their downstream genes, such as CYP3A4, CYP2C9, and P-gp transporters, and they may induce herb–drug interactions. This is an important and emerging field of phytomedicine. The herb–drug interaction of grapefruit is widely known with anti-cancer and anti-infective agents, antilipemic agents, cardiovascular agents, CNS agents, gastrointestinal agents, and immunosuppressants, etc. [90]. The goal of the prior study was to evaluate *Citrus aurantium* L.’s drug–drug interactions (Rutaceae). The herb fructus aurantii (FA), which has been shown to have a number of pharmacological properties, is frequently used in clinics as a digestive and expectorant agent. Rat CYP1A2, CYP3A4, and mRNA expression were all significantly upregulated in comparison to the control group, while CYP2E1 protein expression was significantly downregulated and the corresponding mRNA expression and enzyme activity remained unchanged. In HepG2 cells, CYP1A2 and CYP3A4 mRNA expression was statistically up-regulated in comparison to the control group, but CYP2E1 mRNA expression was neither significantly induced nor inhibited [91]. The current study, which was motivated by the “grapefruit (GF) juice effect”, was carried out to examine the herb–drug interactions (HDIs) of citrus herbs (CHs). The 159 compounds in CHs and a total of 249 components in GF showed excellent potential as active ingredients based on network analysis. In addition, 360 genes related to GF, 422 to CH, and 111 to drug transport and metabolism were gathered, and 25 and 26 overlapping genes were found. Higher levels of naringenin, isopimpinellin, apigenin, sinensetin, and isoimperatorin were found in compound–target networks, and the findings of protein–protein interactions suggested that CYP3A4 and UGT1A1 served as the hub proteins. Through a variety of drug transporters and drug-metabolizing enzymes, conventional medications such as erlotinib, nilotinib, tamoxifen, theophylline, venlafaxine, and verapamil were linked to GF and CHs. Surprisingly, GF and CHs shared 48 potential active substances, with naringenin, tangeretin, nobiletin, and apigenin having the strongest interactions with targets [92]. This study’s goal was to assess the HDI potential of plants that are frequently used as components in BDS and other herbal products. The findings suggest that long-term or excessive use of herbal preparations containing these plants may increase the risk of CYP- and P-gp-mediated HDIs, which could result in unpleasant side effects from the altered pharmacokinetics of concurrently taken medications [93].

## 6. Health Benefits of Bioactive Substances Derived from Citrus Waste

Several secondary metabolites (coumarins, flavonoids, carotenoids, limonoids, phenolic acids, essential oils (EOs), and alkaloids) are found in citrus fruits, as shown in Figure 5. Citrus secondary metabolites have a biological effect on human health, including neuroprotective, anti-inflammatory, anti-oxidative, anti-cancer, and cardiovascular-protective effects [94]. The health benefits of the bioactive compounds in citrus are mentioned in Table 2.

### 6.1. Functional Benefits in the Control of Diabetes

In a research work, a novel preventive beverage/functional food was developed using a variety of citrus fruits with high antioxidant activity and polyphenolic content. The results revealed that the citrus extracts had the potential to treat diabetes, cardiovascular disease, and cancer, as they contained nutraceuticals and bioactive compounds [105]. Citrus flavonoids have the ability to boost insulin release and reactivity, improve glucose tolerance and peripheral glucose uptake, modulate enzyme activity, reduce the levels of insulin resistance and hepatic glucose, and suppress inflammation [71].

### 6.2. Anti-Cancer Properties

Cancer is a leading cause of sudden death around the world. In recent decades, the interest of scientists and researchers has increased in the development of functional food products with excellent anti-cancer properties. Some functional food products have several limits in terms of applicability and effectiveness, and they are frequently linked with significant adverse consequences, which can lower the overall quality of life for patients [106]. In the field of functional foods, the recent discovery of drugs based on natural products that work against cancer is gaining more popularity, with a few demonstrating efficacy and little toxicity in the management and treatment of the cancer-causing process. The peels of citrus fruits and other comparable extracts have demonstrated promising anti-cancer effectiveness due to the presence of high flavonoid content. To conclude, in vivo studies reveal that citrus peel flavonoids, such as citrus peel extracts and nobiletin, have significant anti-tumor properties in the treatment of malignancies of the colon, skin, lung, prostate, and liver cancer. Inflammation, angiogenesis, proliferation, and apoptosis may all be inhibited as part of the therapeutic strategy [107]. The health benefits of citrus fruits have been known for a long time, making them the most often consumed fruits in the Mediterranean diet. Observational studies have shown a correlation between the consumption of flavonoids in one’s diet and a reduced risk of cardiovascular disease and cancer [108]. Citrus fruits appear to be important for human health due to the presence of vitamins, minerals, and fiber content. The anti-inflammatory and antioxidant properties of citrus fruits have been found to be promising for preventing chronic degenerative illnesses, including cancer and atherosclerosis. Citrus fruits provide critical nutrients such as mineral salts, vitamins, and trace elements, as well as a variety of functional molecules that help the body to maintain its natural homeostasis. Citrus flavonoids can be used in the treatment and prevention of atherosclerosis and cancer in humans [109].

### 6.3. Mental Health and Metabolism

As people live longer, age-related disorders such as Alzheimer’s disease have emerged as major health problems throughout the world. Amyloid buildup, oxidative stress, hyperphosphorylation of tau, and neuroinflammation are hallmarks of Alzheimer’s disease. Furthermore, lifestyle-related disorders such as diabetes, dyslipidemia, vascular dysfunction, and obesity have been associated with an increased risk of dementia. Therefore, the interest of scientists and researchers have been increased towards the development of new methods to maintain brain function and prevent dementia in older and pre-clinical individuals. Citrus fruits are rich in polymethoxylated flavones and flavanones that are beneficial for brain function. In this pre-clinical research, results showed that they had neuroprotective properties in dementia models such as Alzheimer’s disease [110]. Similarly, in another clinical and epidemiological study, flavonoids in citrus peels and extracts led to greater cognitive performance and lowered the disease risk of depression and dementia. Results confirmed the citrus peels’ and extracts’ positive impacts on human cognition and associated functions [111].

### 6.4. Citrus’ Role in the Prevention of Oxidative Damage and Cardiovascular Disease

Cardiovascular illnesses are a serious concern worldwide, with many deaths reported each year, and treatment resistance is becoming increasingly common. This area of research needs more attention from scientists and researchers in order to discover new, natural methods to isolate beneficial chemicals from citrus fruits such as lemons, sweet oranges, and grapefruits, which are used in the treatment of hypertension. Citrus extract has the ability to significantly reduce hemorrhages. Citrus fruits contain the flavonoids chrysin, luteolin, and 7-hydroxy flavone, which play a major role in lipoprotein induction in the umbilical vein. Endothelial cells in the human body oxidize low-density lipoprotein (LDL) and create more reactive oxygen species inside the cells. LDL regulates the bioactive chemicals in our bodies. Bioactive chemicals are more effective in controlling hypercholesterolemia and atherosclerosis [112]. Citrus flavonoids, among other qualities, scavenge free radicals, change lipid metabolism, increase glucose tolerance and insulin resistance, promote adipocyte differentiation, alleviate endothelial dysfunction, and reduce inflammation and apoptosis. It has been found that flavonoids present in citrus fruits can lead to better outcomes for cardiovascular health [71].

Effective heart failure self-care has been linked to depression, but more recent research suggests that the connection between the two is more complex than previously thought [113]. In patients with chronic heart failure, insufficient daily intake of citrus fruits was associated with an increased risk of developing depression. Our findings support the theory that consuming citrus fruits daily can help to prevent and manage depression in chronic heart failure patients [114].

### 6.5. Citrus’ Role in the Prevention of Obesity

Obesity is a chronic and life-threatening condition affecting people worldwide. Anti-obesity drugs can have serious side effects, and there is no assurance that these treatments can aid patients in the long term [115]. In the last few decades, complementary and alternative medicines have gained a great deal of attention due to their excellent health properties. Citrus phytochemicals have shown great promise in a variety of methods in treating obesity disorders. Citrus fruits have been identified as the main active biological ingredient that contains higher amounts of flavonoids and p-synephrine. In many research works, it was revealed that citrus fruits have anti-obesity properties through a variety of mechanisms, including energy intake and expenditure control, lipid metabolism modulation, and adipogenesis regulation [116]. They possess anti-obesity characteristics through a range of pathways, some of which include the control of energy intake and expenditure, the modification of lipid metabolism, and the regulation of adipogenesis. In another research study on citrus phytochemicals and their anti-obesity benefits, as well as an update of studies conducted within the past ten years, the active components and mechanisms involved in their anti-obesity action were considered. Citrus phytochemicals have been shown to reduce the risk of developing obesity [117]. According to in vitro and in vivo research, hydroxyl PMFs and polymethoxyflavones present in citrus peels have shown anti-obesity potential. This study investigated the anti-obesity potential of two citrus peel extracts and findings indicated that orange peel extracts reduced fat storage in both in vivo and in vitro analysis and that they should be investigated as a treatment for obesity and overweight [118].

### 6.6. Regulation of Lipoprotein Metabolism

Citrus fruits have been recognized as a rich source of flavonoids, vitamins, carotenoids, and polyphenolic compounds with excellent biological properties. Recent studies have focused on the potential of citrus flavonoids to influence lipid metabolism, along with other metabolic factors relating to metabolic syndrome, heart disorders, type 2 diabetes, and atherosclerosis disorders. Nobiletin, naringenin, tangeretin, and hesperidin are major citroflavonoids that have emerged as promising nutraceutical and therapeutic agents for the treatment of metabolic dysregulation [119]. According to epidemiological studies, the consumption of citrus flavonoid-rich foods is associated with a lower risk of cardiovascular disorders, inflammation, and metabolic disorders. In another clinical research work, it was found that citrus flavonoids are promising antibiotic, antihypertensive, insulin-sensitizing, lipid-lowering, antidiabetic, and anti-inflammatory agents [120]. In animal models, the use of citrus flavonoids as a dietary supplement in mice diets reduced the risk of hepatic steatosis, dyslipidemia, and insulin resistance by inhibiting hepatic fatty acid production and increasing fatty acid oxidation. Citrus flavonoids also reduced inflammation in metabolically essential tissues such as the liver, adipose tissue, kidneys, and aorta. In another recent research work, citrus flavonoids have been shown to help with insulin resistance, dyslipidemia, obesity, hepatic steatosis, and atherosclerosis [121].

### 6.7. Phytochemicals as Antiaging Agents

All living organisms change with aging due to a natural physiological process, and the most exogenous factors influencing the degree of ageing are hormones, smoking, nutrition, and direct and indirect exposure to sunshine. This process produces reactive oxygen species, which are unstable molecules and prevent key enzymes from performing their functions in the cell matrix. Antioxidants are able to neutralize free radicals. Lime plants have natural phytochemical compounds that act as antioxidants and anti-aging [122]. These natural phytochemical compounds play important roles in several biological features and physiological activities. Products derived from citrus fruits and their waste are viewed as sustainable, low-cost sources of chemicals with potential use in the pharmacological, cosmetic, culinary, and nutraceutical fields [123].

## 7. Toxicity of Phytochemicals Extracted from *Citrus* L.

The consumption of plants has been a common practice since the dawn of civilization due to their health benefits, prevention or treatment of diseases, or medicinal purposes. Many plants (including fruits and vegetables) contain phytochemicals that are not necessary nutrients, such as vitamins or minerals, but are frequently consumed or used as herbal remedies or dietary supplements due to their alleged health benefits. The Food and Drug Administration (FDA) does not regulate the majority of phytochemicals in the United States, and little is known about their potential toxicity. The use of phytochemical dietary supplements and herbal remedies has increased in recent years, especially in western nations. In the U.S., 49% of the population (45% of men and 53% of women) reported using supplements [124]. Despite their popularity, the safety and effectiveness of the majority of individual phytochemicals are understood in preventing or treating chronic diseases such as cancer [125]. According to epidemiological studies, many consumers of dietary supplements believe that these ingredients are secure and could serve as a more natural substitute for prescription drugs [126]. In fact, some studies have indicated that phytochemicals, such as resveratrol or epigallocatechin gallate (EGCG), may be helpful in the prevention or treatment of chronic diseases [127,128]. Medicinal plants are a diverse source of bioactive compounds that, depending on the species, plant part, and dosage used, may be beneficial or harmful to human health [129,130]. The idea that medicinal plants and herbs are safe and do not exert any toxic effects on humans has been prevalent for a very long time. However, despite the increased reliance on medicinal plants globally, it has not been acknowledged that these plants can contain either beneficial or harmful bioactive compounds [131]. Some constituents may cause the dysfunction of vital organs (including the kidneys, liver, and stomach, as well as the nervous system) [132]. Toxicity studies are a crucial prerequisite in identifying non-toxic, safe, and pharmacologically active plants correctly. One of the main components of citrus essential oils is limonene, which causes high mortality at increased doses and has favorable effects on the life history traits of adult medflies when they are exposed to it at low sub-lethal doses [133]. One of the most widely consumed citrus fruits is the pomelo (*Citrus grandis* L. Osbeck). The common flavonoids reported to be present in pomelo juice in significant amounts include neohesperidin, hesperidin, naringenin, naringin, and rutin [134,135]. According to reports from Mokbel and Suganuma [136] and Do Ngoc et al. [137], the fruit extract has a number of pharmacological advantages, including anti-obesity, antimicrobial, antioxidant, and antidiabetic properties. Despite having many different uses, there is still a lack of information on the safe dosage, safety, and toxicology profiles of many medicinal plants, including pomelo. *Citrus limon* has a long history of use, but little is known about its toxicity. This study was conducted to assess the juice’s potential acute and subacute toxicity. Although it could be regarded as non-toxic and incredibly safe for consumption, even at concentrations above 80%, *Citrus limon* juice does not exhibit potential hazards that are harmful to experimental animals [138]. Citrus peels contain a large amount of essential oils and are toxic to a variety of insect species. The use of essential oils to control insect pests is also environmentally friendly. A waste product called *Citrus maxima* peel essential oil (CMEO) was described and its potential for controlling insect pests was assessed. Limonene and pinene are the two main terpenoids in CMEO. CMEO demonstrated potential for toxicity-based contact and fumigant-based insect pest control. Additionally, Armigeres subalbatus was more resistant to CMEO’s larvicidal effects than the Culex tritaeniorhynchus and Aedes aegypti species of mosquitoes. A discernible germination inhibition was found in comparison to the control group when the essential oil’s biological safety was not tested on stored seeds [139].

## 8. Conclusions

It is concluded that different parts of citrus fruit waste (peel, seeds, and pomace) contained various phytochemicals. This waste, produced during the processing of citrus fruits, contains numerous essential oils (EOs), ascorbic acid, sugars, carotenoids, flavonoids, dietary fiber, polyphenols, and numerous trace components. These compounds are extracted by using different extraction techniques, including ultrasonic extraction, microwave-assisted extraction, and enzymatic extraction. These extracts are being used in the development of different functional food products. These food products play an important role in health. Polyphenols present in citrus waste foods have a variety of potential health benefits, including antiaging, anti-mutagenic, antidiabetic, anti-carcinogenic, anti-allergenic, antioxidative, anti-inflammatory, neuroprotective, and cardiovascular-protective activity.

## Figures and Tables

**Figure 1 molecules-28-01636-f001:**
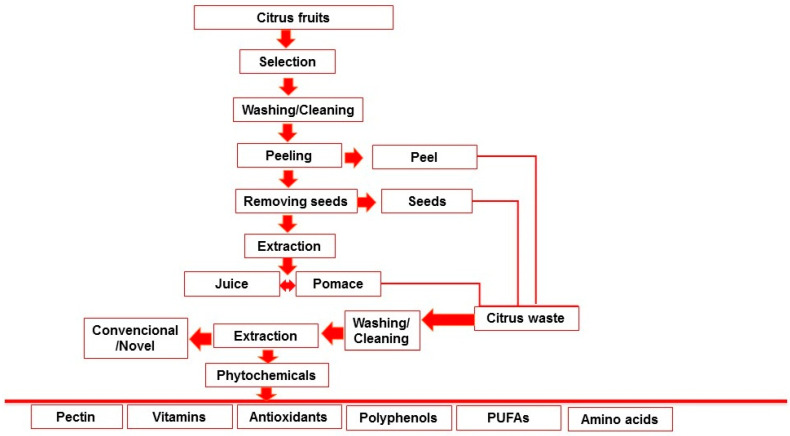
Generation of waste in juice processing.

**Figure 2 molecules-28-01636-f002:**
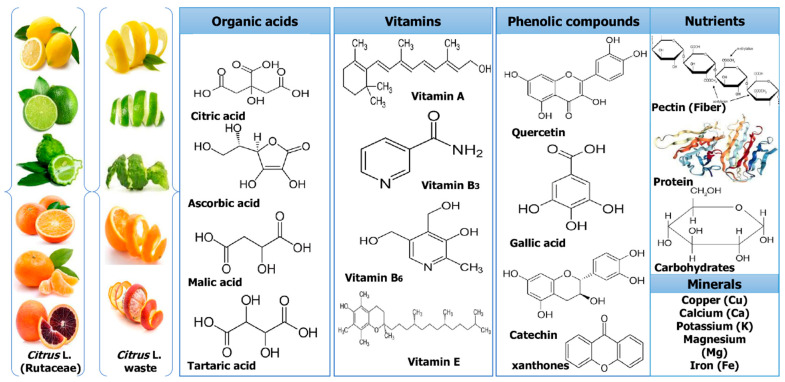
Major phytochemicals of *Citrus* L. waste.

**Figure 3 molecules-28-01636-f003:**
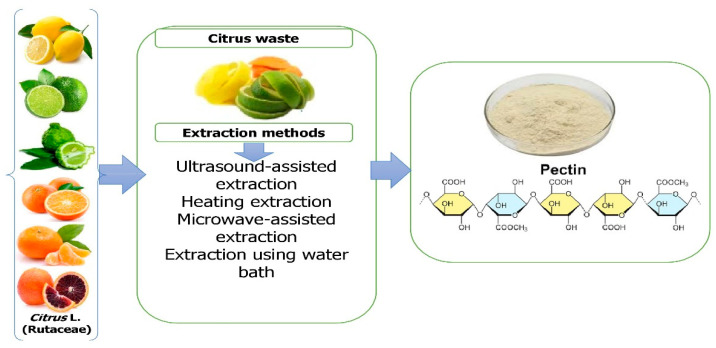
Extraction of pectin from *Citrus* L. waste.

**Figure 4 molecules-28-01636-f004:**
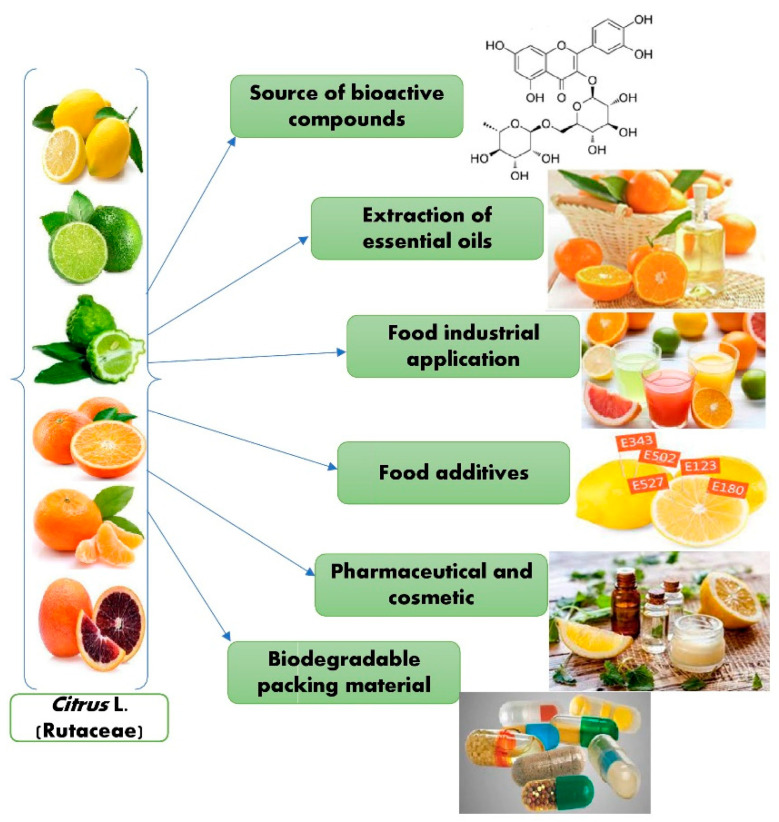
Potential applications of *Citrus* L.

**Figure 5 molecules-28-01636-f005:**
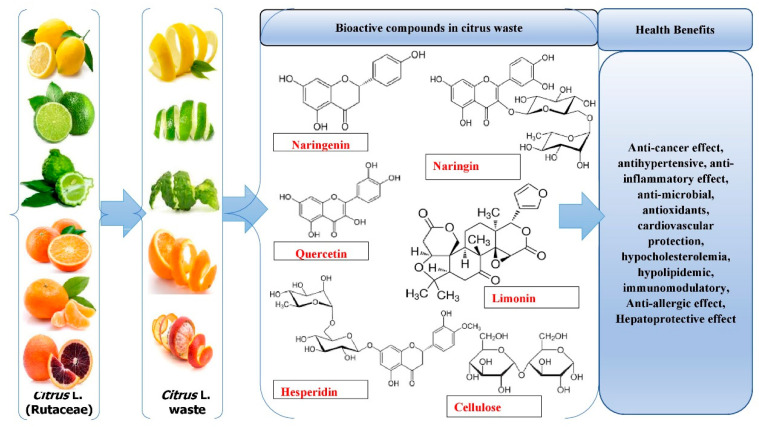
Bioactive compounds in citrus waste and their health benefits.

**Table 1 molecules-28-01636-t001:** Compositions and potential food applications of citrus fruits.

Citrus Source	Part	Composition	Food Application	References
Citron	Peel, pulp	Antioxidants, vitamin C, phenols, flavanols	Carbonated drinks, alcoholic beverages, jams, syrup	[36]
Lemon	Peel	Vitamin C, pectin, phenols, carotenoids	Edible coatings, use as film matrixes	[37,38]
Sweet orange	Peel	Narirutin, naringin, quinic acid, sakuranetin	Enhance vegetable oil oxidative stability, candied orange peel, salad dressings, desserts	[39,40]
Grapefruit	Peel	Naringin, naringenin, flavonoids, phenols	Jam making, sauces, dessert recipes, flavoring of beverages	[32,41]

**Table 2 molecules-28-01636-t002:** Health benefits of bioactive compounds in citrus.

Citrus Source	Scientific Name	Bioactive/Functional Compounds	Diseases	References
Lemon	*Citrus limon*	Eriocitrin, hesperidin, 6,8-di-c-glu-apigenin, quercetin, hesperetin	Antioxidant action, reduce risk of cardiovascular complications, antimicrobial effects	[95]
Citron	*Citrus medica*	Phenolics, flavonones, vitamin C, pectin	Anti-catarrha, anti-hypertensive, antibacterial, antifungal, anti-cancerous	[96]
Pomelo	*Citrus maxima*	Naringin, naringenin, phenols	Antihyperlipidemic properties	[97]
Sweet orange	*Citrus sinensis*	Hesperidin, sinensitin, liminoids, polyphenols	Prevent arteriosclerosis, reduce risk of kidney stones, reduce cholesterol levels, improve stomach ulcer anti-inflammatory effect	[98]
Lime	*Citrus lotifolia*	Luteolin, tangerine, hesperidin	Improve immunity, prevent kidney stones, promote healthy skin	[99,100]
Grapefruit	*Citrus paradisi*	Naringin, narirutin, neoponcirin, quercetin	Antioxidant, anti-inflammatory, and anti-tumor activity; reduces the risk of atherosclerosis; increases bone cell activity	[101,102]
Kumquat	*Citrus japonica*	Polyphenols, essential oils (EOs), vitamin C	Antioxidant activity, liver protection, anti-cancer activity	[103,104]

## Data Availability

Not applicable.

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
