# Peer review of "Citrus Waste as Source of Bioactive Compounds: Extraction and Utilization in Health and Food Industry"

_molecules, 2023, doi:10.3390/molecules28041636_

Round 1
Reviewer 1 Report (Previous Reviewer 1)
This manuscript has been resubmitted, and I still think that could be improved. Include a flowchart indicating the generation of each waste in juice processing. Examples from patents in extraction item could contribute too.
Author Response
Response: We appreciate the reviewer for the valuable suggestions. The flowchart has been added according to suggestion of reviewer.

Reviewer 2 Report (New Reviewer)
1. The quality of Figure 1 is very inferior. At first sight, it appears that the authors have just copied and pasted it. The authors can use Chemdaw for chemical structures.
2. What is the meaning of the two separate pictures in figure 1. Besides, the legend of figure 1, "Chemical composition Citrus L." is improper. It is not a chemical composition. Every plant part, including fruit, contains thousands of phytochemicals, and here, the authors did not take care of mentioning even the name of twenty phytochemicals. Here legend should be "Major active phytochemicals reported from Citrus L waste".
3. The section "essential oils 2.3" the author repeated basic information on essential oil and did not mention the name of any phytochemicals of essential oil. The authors should mention here the phytoconstituent of Citus L essential oils. If a possible species-specific constituent of essential oils will provide. Frame the sentences properly and repeat this process with all major headings.
4. Authors need to follow the rule of the classification system and maintain uniformity throughout the manuscript. Besides, it will be excellent if authors first read about "binomial nomenclature". The corresponding author can at least pay attention towards the manuscript and check basic concepts and rules before submission.
5. The authors have described the biological properties of phytochemicals extracted from Citus L but along with the benefits, the authors need to add a separate heading, "toxicities of phytochemicals extracted from Citus L".
6. Plants, including Citrus L, are used as essential ingredients in the preparation of herbal medicine, dietary supplements, and functional foods. In the manuscript, the authors claimed that phytochemicals of Citrus L waste could be used in formulations of herbal medicine, dietary supplements, and functional foods.
Likewise, the phytochemicals of Citrus L also act as agonists or antagonists with various nuclear receptors such as PXR, CAR and AhR and regulate their downstream genes, such as CYP3A4, CYP2C9 and P-gp transporters and may induce herb-drug interaction. This is an important and emerging field of phytomedicine. The herb-drug interaction of Grapefruit is widely known with anticancer, Anti-infective agents, Antilipemic agents, Cardiovascular agents, CNS agents, Gastrointestinal agents and Immunosuppressants etc. The US Food and Drug Administration (FDA) issues many warnings about the use of grapefruit with medications. The authors could add some information on "Herb-drug Interaction of Citrus L" and for the basic introduction of this section following articles: Am Fam Physician. 2017;96(2):101-107; https://doi.org/10.1159/000334488; https://doi.org/10.1016/j.phymed.2020.153416; https://doi.org/10.1002/fft2.110; https://doi.org/10.1016/j.heliyon.2020.e05357; https://doi.org/10.1016/j.jep.2022.115822; https://doi.org/10.3390/jcm11061567; doi: 10.3390/molecules26082315, doi.org/10.3389/fphar.2012.00069 should be helpful and can be used as references.
Overall, the subscript quality is inferior. Instead of making just publications, senior authors should pay attention and include some recent and significant content that can be helpful to the audience and maintain the Journal's integrity.
Author Response
Reviewer 2
Comments and Suggestions for Authors
- The quality of Figure 1 is very inferior. At first sight, it appears that the authors have just copied and pasted it. The authors can use Chemdaw for chemical structures.
Response: The Figure has been redesigned according to suggestion of reviewer.
- What is the meaning of the two separate pictures in figure 1. Besides, the legend of figure 1, "Chemical composition Citrus L." is improper. It is not a chemical composition. Every plant part, including fruit, contains thousands of phytochemicals, and here, the authors did not take care of mentioning even the name of twenty phytochemicals. Here legend should be "Major active phytochemicals reported from Citrus L waste".
Response: Changes have been made in the revised version
- The section "essential oils 2.3" the author repeated basic information on essential oil and did not mention the name of any phytochemicals of essential oil. The authors should mention here the phytoconstituent of Citus L essential oils. If a possible species-specific constituent of essential oils will provide. Frame the sentences properly and repeat this process with all major headings.
Response: Changes have been made in the revised version
- Authors need to follow the rule of the classification system and maintain uniformity throughout the manuscript. Besides, it will be excellent if authors first read about "binomial nomenclature". The corresponding author can at least pay attention towards the manuscript and check basic concepts and rules before submission.
Response: The manuscript has been improved in the revised version
- The authors have described the biological properties of phytochemicals extracted from Citus L but along with the benefits, the authors need to add a separate heading, "toxicities of phytochemicals extracted from Citus L".
Response: The new section (toxicities of phytochemicals extracted from Citus L) has been added according to suggestion of reviewer.
- Plants, including CitrusL, are used as essential ingredients in the preparation of herbal medicine, dietary supplements, and functional foods. In the manuscript, the authors claimed that phytochemicals of Citrus L waste could be used in formulations of herbal medicine, dietary supplements, and functional foods.
Likewise, the phytochemicals of Citrus L also act as agonists or antagonists with various nuclear receptors such as PXR, CAR and AhR and regulate their downstream genes, such as CYP3A4, CYP2C9 and P-gp transporters and may induce herb-drug interaction. This is an important and emerging field of phytomedicine. The herb-drug interaction of Grapefruit is widely known with anticancer, Anti-infective agents, Antilipemic agents, Cardiovascular agents, CNS agents, Gastrointestinal agents and Immunosuppressants etc. The US Food and Drug Administration (FDA) issues many warnings about the use of grapefruit with medications. The authors could add some information on "Herb-drug Interaction of Citrus L" and for the basic introduction of this section following articles: Am Fam Physician. 2017;96(2):101-107; https://doi.org/10.1159/000334488; https://doi.org/10.1016/j.phymed.2020.153416; https://doi.org/10.1002/fft2.110; https://doi.org/10.1016/j.heliyon.2020.e05357; https://doi.org/10.1016/j.jep.2022.115822; https://doi.org/10.3390/jcm11061567; doi: 10.3390/molecules26082315, doi.org/10.3389/fphar.2012.00069 should be helpful and can be used as references.
Response: The new section (Herb-drug Interaction of Citrus L) has been added according to suggestion of reviewer.
Overall, the subscript quality is inferior. Instead of making just publications, senior authors should pay attention and include some recent and significant content that can be helpful to the audience and maintain the Journal's integrity.
Response: The authors are highly thankful to the respected reviewer for their kind suggestions/comments for improving the manuscript. We tried to revise the manuscript according to the valuable advices.

This manuscript is a resubmission of an earlier submission. The following is a list of the peer review reports and author responses from that submission.
Round 1
Reviewer 1 Report
This manuscript is about citrus wastes however the focus is about the metabolites present in citrus waste. It is missing a section about how these waste are obtained, including a process flowchart, and if there are differences in the processing of different citric fruits. If the authors think it is too much, I suggest to exclude the idea of waste, and concentrate in bioactives in general, no specifying if it is waste or not
Author Response
Comments and Suggestions for Authors
This manuscript is about citrus wastes however the focus is about the metabolites present in citrus waste. It is missing a section about how these waste are obtained, including a process flowchart, and if there are differences in the processing of different citric fruits. If the authors think it is too much, I suggest to exclude the idea of waste, and concentrate in bioactives in general, no specifying if it is waste or not
Response: The authors are highly thankful to the respected reviewer for their kind suggestions/comments for improving the manuscript. We tried to revise the manuscript according to the valuable advices. However, section has been added according to suggestion of reviewer.
Reviewer 2 Report
The study of Citrus is interesting and information about sources of extraction and utilization is promising for the future. However, the review has a number of flaws, and the information provided is too concise. A few comments:
1. "Italic" font is missing (for example Citrus L., Table 2, and other) lines 40, 57, respectively.
2. 49 line the examples of fruits are missing.
3. The citation is missing (line 52; what studies?).
4. Explain abbreviations: CVD, USE, MAE, and others.
5. Information is repeated several times; 69,70 lines and others in all review.
6. Tangerine and mandarin are the same fruit.
7. Unclear pictures, missing information, and there are errors. Chemical composition is much more varied.
8. Write examples of these components (line 104).
9. Look to citation requirements (does the number of citations are written after the author who is cited?).
10. Extraction of pectin is not clear. Add a picture.
11. Does cellulose soluble in water or not? Line 122,123.
12. Scientific names of fruits are not full. Table 2 (example Citrus limon L.).
13. The entire article should be thoroughly reviewed.
14. How can residents contribute to waste reduction and use citrus waste in their households?
15. The conclusions are almost the same as the abstract.
16. More sources are lacking to support the claims. Almost all claims are based on only one study. The information used in the article lacks scientific validity.
Author Response
Comments and Suggestions for Authors
The study of Citrus is interesting and information about sources of extraction and utilization is promising for the future. However, the review has a number of flaws, and the information provided is too concise. A few comments:
Response: We are highly thankful to the respected reviewer for their valuable suggestions for improving the manuscript.
- "Italic" font is missing (for example CitrusL., Table 2, and other) lines 40, 57, respectively.
Response: "Italic" font has been added in the overall sections of manuscript
- 49 line the examples of fruits are missing.
Response: added
- The citation is missing (line 52; what studies?).
Response: added
- Explain abbreviations: CVD, USE, MAE, and others.
Response: The abbreviations have been added in the revised manuscript.
- Information is repeated several times; 69,70 lines and others in all review.
Response: Changes have been made in the revised manuscript
- Tangerine and mandarin are the same fruit.
Response: Changes have been made in the revised manuscript
- Unclear pictures, missing information, and there are errors. Chemical composition is much more varied.
Response: The pictures have been revised.
- Write examples of these components (line 104).
Response: added
- Look to citation requirements (does the number of citations are written after the author who is cited?).
Response: Changes have been made in the revised manuscript
- Extraction of pectin is not clear. Add a picture.
Response: The picture has been added according to suggestion of reviewer.
- Does cellulose soluble in water or not? Line 122,123.
Response: cellulose is not soluble in water. However, correction has been made in the revised version.
- Scientific names of fruits are not full. Table 2 (exampleCitrus limonL.).
Response: added
- The entire article should be thoroughly reviewed.
Response: reviewed
- How can residents contribute to waste reduction and use citrus waste in their households?
Response: Information has been added in the revised version.
- The conclusions are almost the same as the abstract.
Response: Conclusion has been rewritten
- More sources are lacking to support the claims. Almost all claims are based on only one study. The information used in the article lacks scientific validity.
Response: Changes have been made. However, we tried to improve the manuscript
Round 2
Reviewer 1 Report
The authors answered my question, so it can be approved
Reviewer 2 Report
Thanks for the corrections. The article has been significantly corrected; however, there are many inaccuracies.
English language must be carefully revised.
Many inaccuracies are left in the text.
In my opinion, the manuscript should be rejected. A few suggestions for further fixes.
1. The composition shown in Figure 1 is not complete. The first version of the article listed minerals, fiber, protein, etc. Maybe you could add to this picture all the compounds that Citrus L. accumulates.
2. The scientific names of the plants are given incorrectly (table 2). Use the Taxonomy browser.
3. How about pesticides and other chemical compounds on the citrus surface which could be dangerous to health? To use citrus waste they have to be washed and otherwise processed.
The article lacks a broader approach; many ideas are repeated several times. There is a lot of text, but the main idea, the benefits of utilization, and safety are not revealed.